# Integrating prediction errors at two time scales permits rapid recalibration of speech sound categories

**Itsaso Olasagasti\*, Anne-Lise Giraud**

Department of Basic Neuroscience, University of Geneva, Geneva, Switzerland

**Abstract** Speech perception presumably arises from internal models of how specific sensory features are associated with speech sounds. These features change constantly (e.g. different speakers, articulation modes etc.), and listeners need to recalibrate their internal models by appropriately weighing new versus old evidence. Models of speech recalibration classically ignore this volatility. The effect of volatility in tasks where sensory cues were associated with arbitrary experimenter-defined categories were well described by models that continuously adapt the learning rate while keeping a single representation of the category. Using neurocomputational modelling we show that recalibration of *natural* speech sound categories is better described by representing the latter at different time scales. We illustrate our proposal by modeling fast recalibration of speech sounds after experiencing the McGurk effect. We propose that working representations of speech categories are driven both by their current environment and their long-term memory representations.

**\*For correspondence:**
itsaso.olasagasti@gmail.com

**Competing interests:** The authors declare that no competing interests exist.

## Introduction

The way the brain processes sensory information to represent the perceived world is flexible and varies depending on changes in the stimulus landscape. Neurocognitive adaptation to varying stimuli can be driven by an explicit external feedback signal, but might also take place with simple passive exposure to a changing stimulus environment via implicit statistical learning (*Saffran et al., 1996*; *Gilbert et al., 2001*; *Barascud et al., 2016*; *Schwiedrzik et al., 2014*). In the domain of speech perception, neural representations of sound categories are susceptible to stimulus-driven recalibration. Typically, the perception of unclear or ambiguous speech stimuli that have previously been disambiguated by context (*McQueen et al., 2006*; *Clarke and Luce, 2005*) or by a concurrent visual stimulus (*Bertelson et al., 2003*; *Vroomen et al., 2007*) is biased by the disambiguating percept. Even simple exposure to novel statistics e.g., a variation in the spread of sensory features characteristic of stop consonants, quickly results in a modified slope in psychometric functions, and changes the way listeners classify stimuli (*Clayards et al., 2008*). Interestingly, acoustic representations are also modified after altered auditory feedback during production (*Nasir and Ostry, 2009*; *Lametti et al., 2014*; *Patri et al., 2018*). These observations illustrate that speech sound categories remain largely plastic in adulthood.

Using two-alternative forced choice tasks, studies have shown that changes in acoustic speech categories can be induced by input from the visual modality (e.g. *Bertelson et al., 2003*). Reciprocally, acoustic information can also disambiguate lipreading (*Baart and Vroomen, 2010*), resulting in measurable categorization aftereffects. While such effects can be observed after repeated exposure to the adapting stimuli, recalibration can also occur very rapidly, with effects being observable after a single exposure (*Vroomen et al., 2007*). This fast and dynamic process has been modelled as incremental Bayesian updating (*Kleinschmidt and Jaeger, 2015*) where internal perceptual categories track the stimulus statistics. According to this model, when listeners are confronted with altered

**eLife digest** People can distinguish words or syllables even though they may sound different with every speaker. This striking ability reflects the fact that our brain is continually modifying the way we recognise and interpret the spoken word based on what we have heard before, by comparing past experience with the most recent one to update expectations. This phenomenon also occurs in the McGurk effect: an auditory illusion in which someone hears one syllable but sees a person saying another syllable and ends up perceiving a third distinct sound.

Abstract models, which provide a functional rather than a mechanistic description of what the brain does, can test how humans use expectations and prior knowledge to interpret the information delivered by the senses at any given moment. Olasagasti and Giraud have now built an abstract model of how brains recalibrate perception of natural speech sounds. By fitting the model with existing experimental data using the McGurk effect, the results suggest that, rather than using a single sound representation that is adjusted with each sensory experience, the brain recalibrates sounds at two different timescales.

Over and above slow "procedural" learning, the findings show that there is also rapid recalibration of how different sounds are interpreted. This working representation of speech enables adaptation to changing or noisy environments and illustrates that the process is far more dynamic and flexible than previously thought.

versions of known speech categories, the perceived category representation is updated to become more consistent with the actual features of the stimulus. The resulting recalibration weighs all evidence equally, disregarding their recency. The model hence successfully describes perceptual changes observed when listeners are confronted with repeated presentations of a single modified version of a speech sound. However, it cannot appropriately deal with intrinsically changing environments, in which sensory cues quickly become obsolete. Real moment to moment physical changes in the environment are referred to as 'volatility' to distinguish them from the trial-by-trial response variability observed in a fixed environment. Inference in volatile environments has been studied mostly in relation to decision making tasks in which participants can use an explicit feedback to keep track of the varying statistics of arbitrary cue-reward associations (e.g. *Behrens et al., 2007*) or arbitrarily defined categories (e.g. *Summerfield et al., 2011*). These studies suggest that humans are able to adjust their learning rate to the volatility in the stimulus set, with faster learning rates (implying a stronger devaluation of recent past evidence) in more volatile environments. This has led to normative models focussing on the online estimation of volatility, in which task-relevant features are represented at a single variable time scale (*Behrens et al., 2007*; *Mathys et al., 2011*).

The notion of variable learning rates likely also applies to speech processing. However, we propose that when recalibrating natural speech categories, a normative model should additionally take into account that these categories may themselves change at different time scales. For example, transient acoustic changes within a given speech sound category, such as those coming from a new speaker, must not interfere with the long-term representation of that category that should be invariant for example to speakers. We therefore hypothesize that speech sound categories could be represented by more than a single varying timescale.

Although speech category recalibration has not been systematically studied in variable environments, Lüttke and collaborators (*Lüttke et al., 2016a*; *Lüttke et al., 2018*) found evidence for recalibration in an experiment that included audio-visual McGurk stimuli shown without an explicit adapting condition. The first study involved six different vowel/consonant/vowel stimuli presented in random order, and recalibration was observed even when acoustic stimuli were not ambiguous (e.g. the /aba/ sound in the McGurk trials). The McGurk effect (the fact that an acoustic stimulus /aba/is mostly perceived as an illusory/ada/ when presented with the video of a speaker producing /aga/) was powerful enough to yield observable adaptive effects across consecutive trials. Specifically, the probability of an acoustic/aba/ to be categorized as/ada/, was higher when the trial was preceded by an audio-visual McGurk fusion. Recalibration effects do not generalize to all phonetic contrasts/ categories (*Reinisch et al., 2014*). After participants had recalibrated acoustic sounds in the /aba /- / ada/continuum (on the basis of acoustic formant transitions), recalibration was neither present for /

ibi /- /idi/ (cued by burst and frication), nor for /ama/- /ana/or to /ubu/- /udu/continua (both cued by formant transitions). Likewise, in a word recognition task, participants were able to keep different F0/VOT (fundamental frequency/voice onset time) correlation statistics for different places of articulation (*Idemaru and Holt, 2014*). Based on this failure to generalize, we hypothesized that very short-term changes can modify the internal mapping between sensory features and sublexical speech categories rather than phonemic categories.

To test this hypothesis, we simulated Lüttke et al.'s experiment (*Lüttke et al., 2016a*), using an audiovisual integration model based on hierarchical Bayesian inference. The model was a version of a previous model of the McGurk effect (*Olasagasti et al., 2015*) that further included an adaptation mechanism using residual prediction errors to update internal representations associated with the perceived category. The model divides the process in two steps: perceptual inference and internal model recalibration. During perceptual inference the model takes the sensory input and infers a perceived speech category, by choosing the category that minimizes sensory prediction error. However, 'residual' prediction errors might remain following perceptual inference. This is typically the case after McGurk fusion; since the best explanation for the multisensory input, 'ada', is neither the audio / aba/ nor the visual /aga/, 'residual' prediction errors remain in both acoustic and visual modalities.

The best match to the experimental results described above (*Lüttke et al., 2016a*) was obtained when we considered that adaptation included two different time scales, resulting in 1) a transient effect leading to recalibration towards the most recently presented stimulus features, decaying towards 2) a longer-lasting representation corresponding to a mapping between category and stimulus features determined within a longer time span.

Overall, these findings are consistent with theories that posit that the brain continuously recalibrates generative (forward) models to maintain self-consistency (e.g., *Friston et al., 2010*), and offers a neuro-computationally plausible implementation to resolve cognitive conflicts, which can sometimes appear as irrational behaviors, such as in post-choice re-evaluation of alternatives (*Coppin et al., 2010*; *Izuma et al., 2010*; *Colosio et al., 2017*; *Otten et al., 2017*).

## Results

### Simulation of the perceptual decision process

In a re-assessment of an existing dataset (*Lüttke et al., 2016a*) selected participants with high percentage of fused percepts for McGurk stimuli. When presented with acoustic /aba/ together with a video of a speaker articulating /aga/ the most frequent percept was /ada/; we refer to these as fused McGurk trials. These listeners were combining information from acoustic (A) and visual (V) modalities since the same /aba/ acoustic token was correctly categorized as /aba/ when presented alone. After a fused McGurk trial, participants showed recalibration. They classified acoustic only /aba/ stimuli as 'ada' more frequently (29% 'ada' percepts) than when the acoustic only /aba/was presented after any other stimulus (16% 'ada' percepts).

Our goal was to compare generative models that interpret sensory input and continuously recalibrate themselves to best match the incoming input. Unlike many other studies of speech recalibration, the stimulus generating recalibration in Lüttke et al. (the McGurk stimulus) was not presented alone, but as part of a set of six stimuli that were presented in random order, thus making transient effects detectable. The assessment involved three acoustic only stimuli with sounds corresponding to /aba/, /ada/ or/aga/; and three audiovisual stimuli – congruent /aba/, congruent /ada/, and the McGurk stimulus (acoustic /aba/with video of /aga/).

We simulated Lüttke et al.'s experiment by using a generative model relating the three possible speech categories (/aba/, /ada/ and /aga/) to the sensory input. We characterized sensory input with a visual feature, the amplitude of lip closure during the transition between the two vowels ($s_V$); and an acoustic feature, the amplitude of the 2$^{nd}$ formant transition ($s_A$). We use 'A' to refer to quantities related to the acoustic feature and 'V' to quantities related to the visual feature.

The internal generative model that characterizes the participant, described in detail in the methods section, generates sensory inputs ($s_A$ and $s_V$) for the congruent versions of each of the three possible categories (k = /aba/, /ada/, /aga/). The model has a representation for each congruent category based on a Gaussian distribution in a two-dimensional feature space, itself a product of two univariate Gaussian distributions centered at ($\theta_{k,V}$ $\theta_{k,A}$) and with standard deviations ($\sigma_{k,V}$, $\sigma_{k,A}$)

for tokens k = {/aba/,/ada/,/aga/} (*Figure 1*, right panel). The model assumes that given a speech token k, $2^{nd}$ formant amplitude $C_A$ and lip closure amplitude $C_V$ for each individual trial are chosen from the corresponding Gaussian distribution (*Figure 1*, right panel). Once values for $C_A$ and $C_V$ have been determined, sensory lip closure and $2^{nd}$ formant transitions are obtained by adding sensory noise (parameterized by $\sigma_V$ and $\sigma_A$), to obtain the sensory input: $s_A$ and $s_V$. During inference, the model is inverted and provides the posterior probability of a token given the noisy sensory input $p(k|s_V,s_A)$.

In both acoustic and audio-visual trials the listener was asked to report the perceived acoustic stimulus in a three-alternative forced choice task. To simulate the listener's choice, we calculated the probability of the acoustic token given the stimulus. In our notation, this probability is expressed by $p(k_A|s_V,s_A)$ for audiovisual stimuli and by $p(k_A|s_A)$ for unimodal acoustic stimuli (see Materials and Methods for details). The percept at a single trial was determined by choosing the category that maximizes the posterior.

The model qualitatively reproduces the average performance across participants in the task. We chose parameters that elicit a very high rate of McGurk percepts (*Figure 2*, middle panel of the bottom row) and assumed that listeners were always integrating the two sensory streams.

## Simulation of the recalibration process

After the model has made the perceptual decision, it is recalibrated. After each trial, we changed the generative model's location parameters associated with the perceived category ($\theta_{k,V}$ $\theta_{k,A}$), which represent the categories through their expected sensory feature values in each modality. This was done for both the acoustic and visual parameters after an audio-visual trial, and for the acoustic parameter after an acoustic trial. We assumed that this happens for every trial as part of a monitoring process that assesses how well the internal model matches sensory inputs. The changes are thus driven by residual sensory prediction error, the difference between the expected and observed values for the modulation amplitudes in each modality.

When listeners consistently reported the fused percept 'ada' when confronted with a video of /aga/ and the sound of /aba/, the presence of the visual stream modified the acoustic percept from 'aba' to 'ada'. Given that the acoustic input did not correspond to the one that was most expected from the perceived token, there was a systematic residual sensory prediction error. Since this residual prediction error was used as a signal to drive the model's adaptation, the /ada/ representation moved towards the McGurk stimulus parameters after a fused percept (*Figure 3A*).

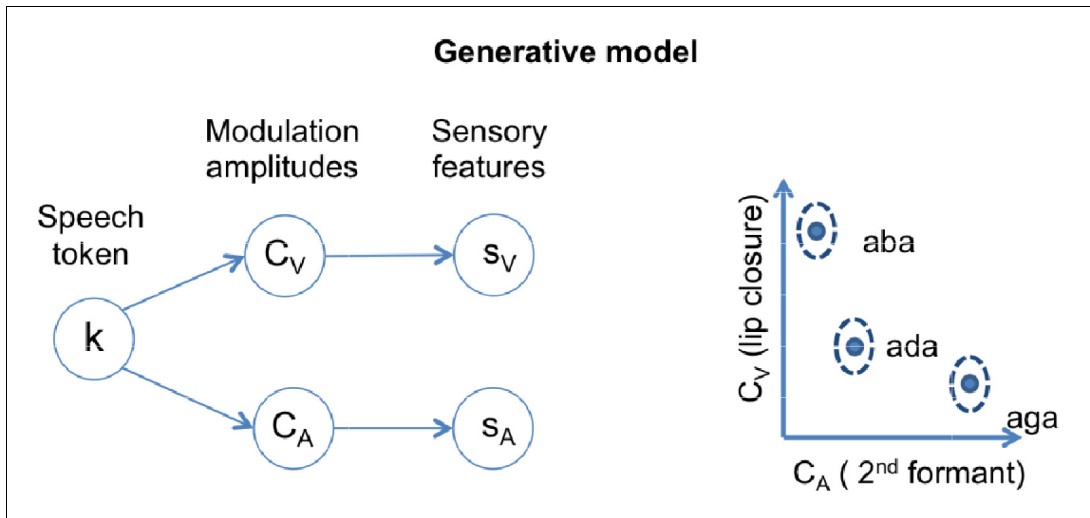

**Figure 1.** Schematics of the generative model. In a given trial, a speech token 'k' determines the amplitudes of degree of lip closure ($C_V$) and magnitude of second formant deflection ($C_A$) by sampling from the appropriate Gaussian distribution. The distributions corresponding to each speech token 'k' are represented in the two-dimensional feature space on the right panel. The model also includes sensory noise to account for how these features appear at the sensory periphery ($s_V$ and $s_A$).

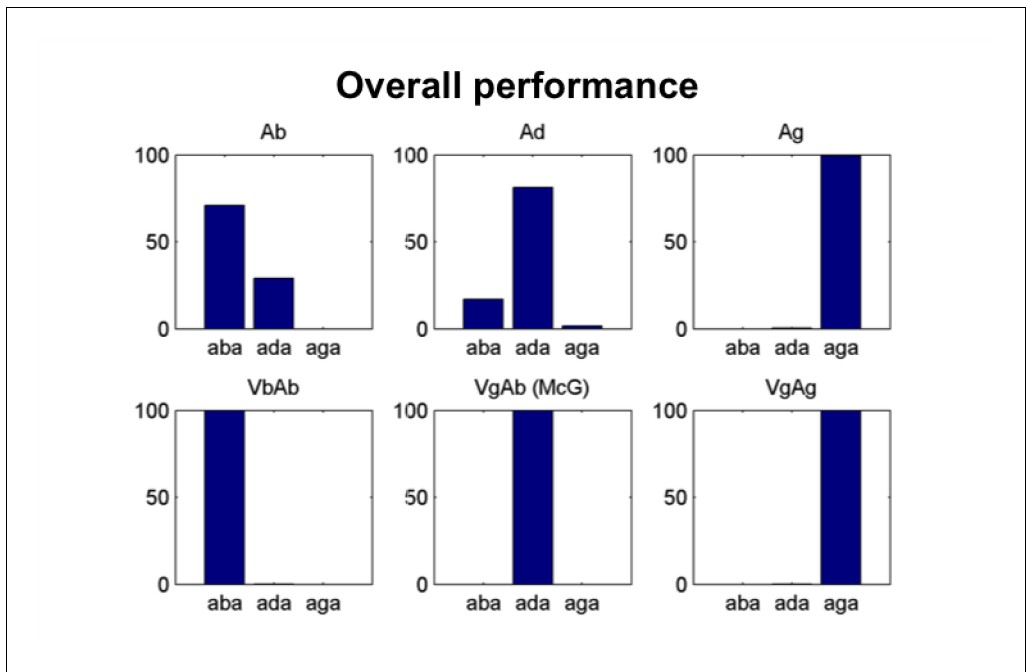

**Figure 2.** Model's overall performance. Simulation of the *Lüttke et al. (2016a)* experiment. Model classification across all trials. Each subpanel shows the percentage of /aba/, /ada/ and /aga/ percepts corresponding to each of the six conditions (Ab: acoustic only /aba/; Ad: acoustic only /ada/; Ag: acoustic only /aga/; VbAb: congruent audiovisual /aba/; VgAb incongruent McGurk stimuli with visual /aga/ and acoustic /aba/; VgAg: congruent /aga/). Congruent and acoustic only stimuli are categorized with a high degree of accuracy and McGurk trials are consistently fused, that is, perceived as /ada/. We reproduce the experimental paradigm consisting of six types of stimuli presented in pseudo-random order; three non-ambiguous acoustic only tokens:/aba/, /ada/ and /aga/, and three audiovisual stimuli: congruent /aba/, incongruent visual /aga/with acoustic /aba/(McGurk stimuli), and congruent /aga/.

In the model, the residual prediction error occurs in the transformation from token identity to predicted modulation of the acoustic feature ($C_A$). Sensory evidence drives estimated $C_A$ towards the experimentally presented value: /aba/ for McGurk stimuli. Thus, when the percept is /ada/, there is a mismatch between the top-down prediction as determined by the top-down component $p(C_A|k)$ that drives $C_A$ towards $\theta_{/ada/,A}$, and the actual value determined by the bottom-up component. This is evident in the following expression

$$C_{k,A} = \frac{\sigma_A^2}{\sigma_A^2 + \sigma_{k,A}^2} \theta_{k,A} + \frac{\sigma_{k,A}^2}{\sigma_A^2 + \sigma_{k,A}^2} S_A \qquad (1)$$

with the first term reflecting the prior expectation for category 'k' and the second reflecting the sensory evidence.

The expression can be rewritten to make the prediction error explicit.

$$C_{k,A} = \theta_{k,A} + \frac{\sigma_{k,A}^2}{\sigma_A^2 + \sigma_{k,A}^2} (S_A - \theta_{k,A}) \qquad (2)$$

This highlights how the listener's estimate of the modulation comes from combining the prediction from the category (first term) and the weighted residual prediction error (in the second term).

To minimize residual prediction error we consider that the participant recalibrates its generative model, which changes $\theta_{k,V}$ and $\theta_{k,A}$ towards $s_V$ and $s_A$. If the stimuli are chosen with parameters 'adapted' to the listener, as we do, $s_V \sim \theta_{stim,V}$ and $s_A \sim \theta_{stim,A}$.

To drive recalibration we considered three different update rules; one derived from the Bayesian model used by *Kleinschmidt and Jaeger (2015)*, which assumes a stable environment and two empirically motivated rules. As a control we also simulated the experiment with no parameter

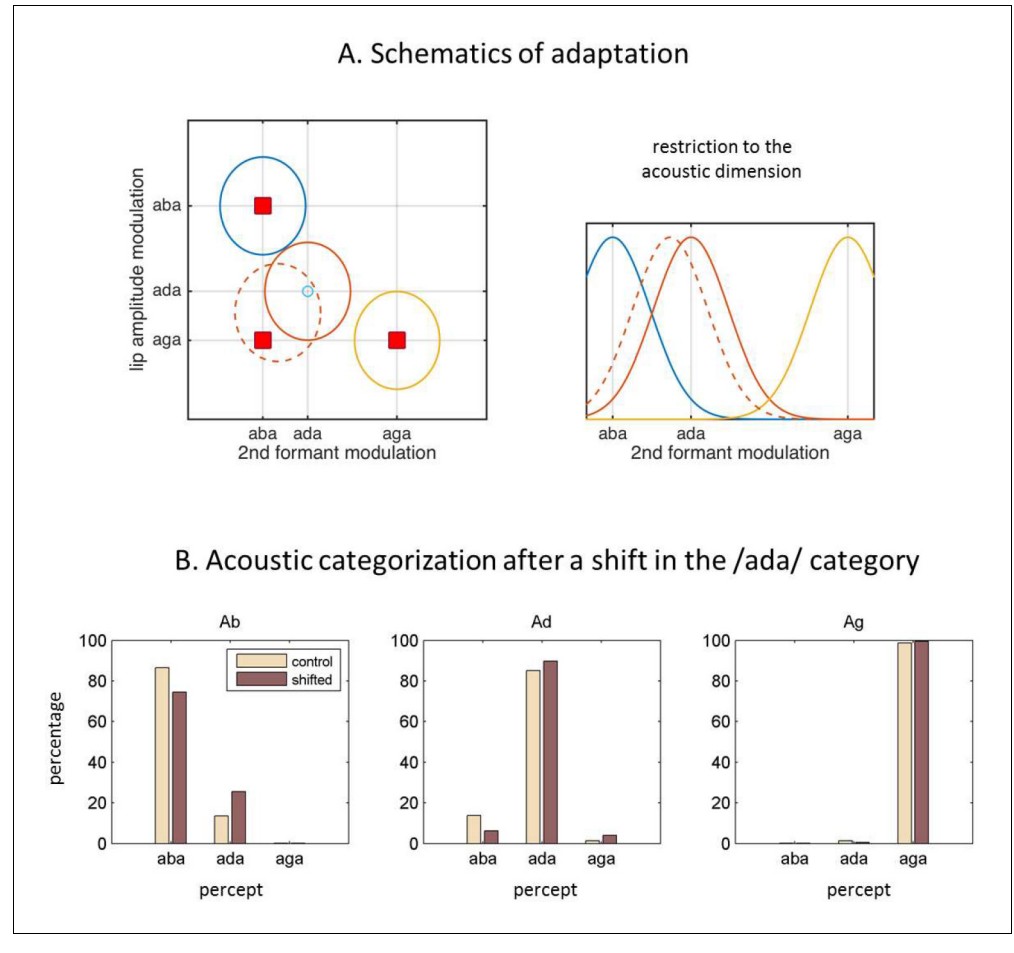

**Figure 3.** Internal model adaptation. (**A**) Speech tokens are represented in a multimodal feature space here represented by two main dimensions. Each ellipse stands for the internal representation of each congruent category ('aba' in blue, 'ada' in red, 'aga' in yellow). The red squares show the location of the audiovisual stimuli in the 2D feature space. They represent congruent /aba/(top left), congruent /aga/(bottom right), and McGurk stimuli (bottom left). When McGurk stimuli are repeatedly perceived as /ada/, the /ada/ representation (in solid red) is modified in such a way that it 'moves' (dashed red) towards the presented McGurk stimulus (visual /aga/ with acoustic /aba/) and therefore should affect the processing of subsequent sensory input. The right panel illustrates how the acoustic representation for /ada/ has shifted towards that of /aba/. (**B**) The effects of the shift in the internal representation on the categorization of the purely acoustic /aba/(Ab), /ada/(Ad) and /aga/(Ag) sounds. Each panel shows the percentage of /aba/, /ada/ and /aga/ percepts for the 'control' representations (solid lines) and the representations with the recalibrated /ada/(dashed line). As in *Lüttke et al. (2016a)*, the biggest effect is observed when categorizing the /aba/ sounds.

updates. For each recalibration model and parameter value set, we run the experiment 100 times, therefore simulating 100 different listeners that share the same perceptual model. Each of the 100 simulated listeners was presented with a different random presentation of the six stimulus types; each presented 69 times (as in the original paper). This gives a total of 414 trials per listener.

To compare with the results from Lüttke et al., who considered 27 participants, we randomly sampled groups of 27 from the 100 simulated listeners to obtain an empirical sampling distribution for the quantities of interest. We focus on the 'McGurk contrast': proportion of /aba/ sounds reported as 'ada' 29% when preceded by a fused McGurk trial versus 16% when preceded by other stimuli. We will also consider the '/ada/ contrast': the difference in the proportion of purely acoustic /aba/ categorized as 'ada' when the preceding trial was a correctly categorized /ada/ sound (17%) versus other stimuli (15%) (percentages correspond to the values reported in *Lüttke et al., 2016a*).

Although we did not perform an exhaustive parameter search, we did repeat the simulations for 20 different values of perceptual model parameters and we also varied recalibration parameter values for each update rule (details in Methods).

Below, for each recalibration model, we report results based on the parameter values that led to the best fit to the 'McGurk contrast'.

## Model without recalibration

As a control we simulated the experiment with no recalibration. For the simulation with the closest fit to the McGurk contrast, the percentage of acoustic /aba/ categorized as 'ada' was 14% after fused McGurk stimuli and 15% after the control stimuli (Wilcoxon signed-rank test p=0.6, *Figure 4A*). The 95% CI for the difference between trials preceded or not by a fused McGurk stimulus was [−6.48, 5.25]%. For the /ada/ contrast, the percentage of acoustic /aba/ categorized as 'ada' was 14% after correctly identified /ada/ sounds and 19% after the control stimuli (Wilcoxon signed-rank test p=0.5). The 95% CI for the difference between the two conditions was [−8.24, 4.21]%.

## Bayesian updating

We first considered the same updating principle as in *Kleinschmidt and Jaeger (2015)* to model changes in speech sound categorization after exposure to adapting stimuli. After each trial the

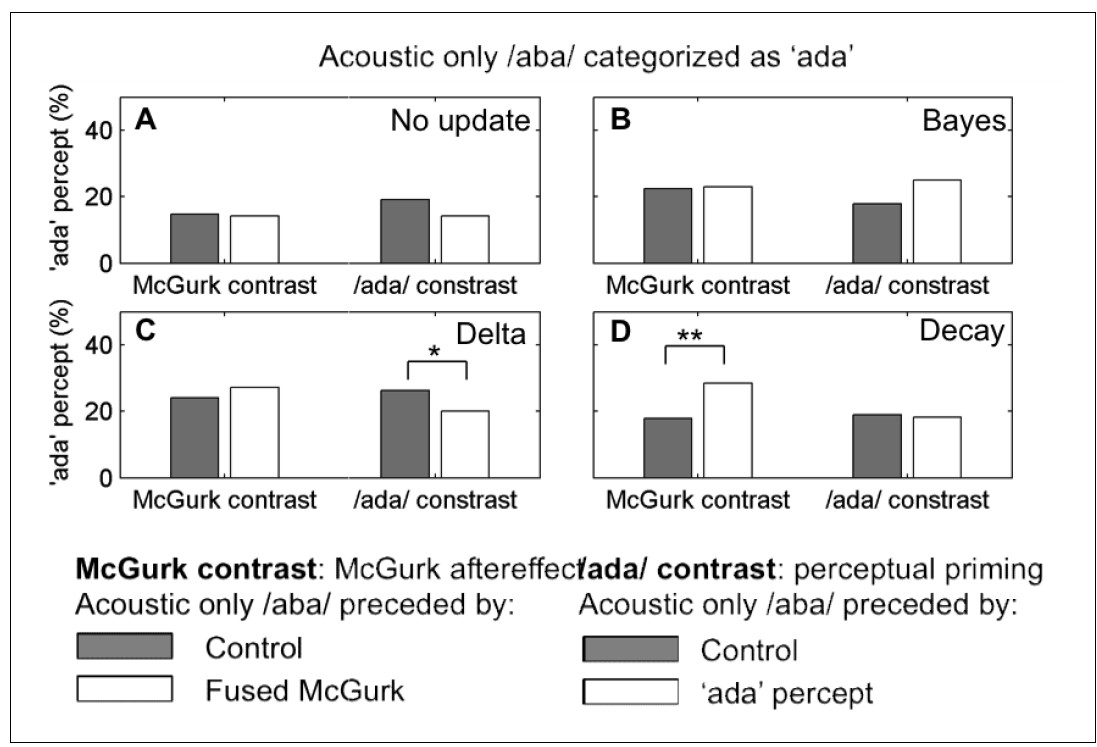

**Figure 4.** Cumulative and transient update rules. 'ada' percepts in response to acoustic /aba/ stimulation. We show two contrasts. The McGurk contrast compares the percentage of 'ada' responses when acoustic /aba/ is preceded by control stimuli (acoustic /aba/ and /aga/, congruent /aba/ and /aga/) versus by fused McGurk trials. The /ada/ contrast refers to acoustic /aba/ preceded by control stimuli (acoustic /aba/ and /aga/) versus acoustic trials correctly perceived as 'ada'. The four panels show the simulation results for the parameters that led to the closest fit to the McGurk contrast reported by *Lüttke et al. (2016a)* for four different update rules as indicated in the insets: (**A**) control, no update, (**B**) the standard Bayesian updates, (**C**) the constant delta rule, and (**D**) Decay, the update rule that assumes that recalibration occurs at two time scales. Both the standard Bayes (**B**) and the constant delta rule (**C**) lead to changes in internal representations that are reflected in the overall increase in 'ada' percepts (with respect to the control, no update model on panel **A**) however, it did not translate into significant effects specific to the next trial. The model assuming two time scales does reproduce the effect of a fused McGurk on the next trial (McGurk contrast). Only the McGurk contrast for the two time scale recalibration model (**D**, left) was significant (\*\*p=0.0003). All other p values were greater than 0.05, except (\*, p=0.03, **C** right).

generative model updates the parameters by considering their probability given the sensory input and the categorization $p(\theta|k\, s_V\, s_A)$, leading to sequential Bayesian updating (*Equation 6* in the methods section). The closest fit to the McGurk contrast was obtained for simulations with $\kappa_{k,f,0} = 1$ and $v_{k,f,0} = 1$. The percentage of /ada/ responses to acoustic only /aba/ stimuli was 23% after fused McGurk stimuli and 22% after control stimuli. This difference was not significant (Wilcoxon signed-rank test p=0.7, *Figure 4B*). The 95% confidence interval for the difference in medians between the two conditions was [−6.14, 7.97]%, thus failing to reproduce the effect of interest. This might be due to the fact that Bayesian update rules have the form of a delta rule with a decreasing learning rate. As a consequence, the magnitude of changes in the categories diminishes as the experiment progresses and all stimuli end up being related to the same internal model. The updates did lead to observable effects; there was an overall increase of /ada/ responses to acoustic /aba/(24% vs. the 14% of the control experiment without parameter updates). The resulting changes in model parameters are expected to induce an after-effect, that is, the point of subjective equivalence in an /aba /- /ada/ acoustic continuum should be shifted in the direction of /aba/.

## Constant delta rule

The Bayesian update rule used above assumes that the parameters are constant in time and that therefore all samples have equal value, whether they are old or recent. This is equivalent to a delta rule with a learning rate tending to zero. We therefore considered a rule with a constant learning rate, which allows for updates of similar magnitude over the whole experiment. The model's expected modulation for the perceived category was recalibrated according to:

$$\Delta\theta_{k,f} = 0.2\,(s_f - \theta_k)\,p(k|s_A,s_V) \ \mathrm{only\, for\, k = percept}$$

where *f* indexes the feature (*f = A*, acoustic feature; *f = V*, visual feature). As *Figure 4C* shows, the percentage of acoustic /aba/ categorized as 'ada' was not significantly higher when the preceding trial was a fused McGurk trial compared with any other stimulus (27% vs. 24%, Wilcoxon signed-rank test, p = 0.08 *Figure 4C*; median difference at 95% CI [−0.14 12]%). Note that in this case too, the proportion of /ada/ responses for acoustic /aba/ inputs is increased compared with simulations run without any adaptation (*Figure 4A*).

Although the learning rate is constant, which means that recalibration magnitude does not necessarily decrease during the experiment, recalibration does not decay across trials. As a result, for trials between consecutive /ada/ percepts, stimuli experience a similar /ada/ category and the simulations do not lead to a significant difference in the classification of acoustic /aba/ whether preceded or not by a fused McGurk trial.

## Hierarchical updates with intrinsic decay

We also tested an alternative update rule that was expected to better reflect how changes occur in the environment. We considered that they might occur hierarchically, with just two levels in a first approximation, corresponding to keeping 'running averages' over different time scales, enabling sensitivity to fast changes without erasing longer-lasting trends.

We considered two sets of hierarchically related variables associated with a single category: $\theta_{k,f}$ (fast) and $\mu_{k,f}$ (slow). The faster decaying one, $\theta_{k,f}$, is driven by both sensory prediction error and the more slowly changing variable, $\mu_{k,f}$ (more details can be found in Materials and Methods). This slowly changing and decaying variable, $\mu_{k,f}$, keeps a representation based on a longer term 'average' over sensory evidence. In the limit, $\mu_{k,f}$ is constant; and this is what we consider here for illustrative purposes. Thus the update rules include an instant change in the fast variable due to the sensory prediction error in the perceived category plus a decay term toward the slower variable for every category. The instant change corresponds to the traditional update after an observation; the decay reflects the transient character of this update.

The results in *Figure 4D* were run with the following parameters:

$$\Delta\theta_{k,f} = 0.4\,(s_f - \theta_k)\,p(k|s_A,s_V) \ \mathrm{only\, for\, k = percept}$$

$$\Delta\theta_{k,f} = 0.14\,(\mu_{k,f} - \theta_{k,f}) \ \mathrm{for\, all\, k}$$

$$\Delta\mu_{\mathrm{k,f}} = 0 \;\; \mathrm{for\,all\,k}$$

where subscript f indexes the feature (f = A for acoustic, f = V for visual). All categories decay toward the corresponding long-term stable values ($\mu_{k,A}$, $\mu_{k,V}$) in the inter-trial interval. By comparing the decay contribution $\Delta\theta_{k,f} = 0.14\,(\mu_{k,f} - \theta_{k,f})$ with the update expression for a quantity with decay time constant $\tau$ in an interval $\Delta t$ (here $\Delta t$ = 5 s, the interval between consecutive trials) $\Delta\theta_{k,f} = \Delta t/\tau\,(\mu_{k,f} - \theta_{k,f})$, we can derive a rough estimate for the decay time constant $\tau \sim 5/0.14$ s $\sim 35$ s.

The percentage of acoustic /aba/categorized as 'ada' after control trials was 18% vs. 29% after fused McGurk (Wilcoxon ranked-signed test, p=0.0004), the median difference being 95% CI: [5.8, 18.0]%. Therefore two effects can be observed; the overall increase in acoustic /aba/ categorized as /ada/ and the rapid recalibration effect reflected in the specific increase observed when acoustic /aba/ was preceded by a fused McGurk trial.

### Update rule comparison

We have modelled recalibration as the continuous updating of the model parameters that represent each of the speech categories used to guide perceptual decisions, in particular the expected values of sensory features associated with each category $\theta_{k,V}$, $\theta_{k,A}$ (k indexing the category). With this approach, the ideal adapter Bayesian account turned out to be incompatible with the experimental findings, due to the erroneous underlying assumption of a stable environment. Because the model assumes that all the sensory observations are derived from exactly the same non-changing distributions, past observations do not lose validity with time. As a result, the location estimate corresponds to the running average of the feature values in the stimuli that have been associated with each category in the course of the experiment. As the occurrence of a perceived category increases, the size of recalibration decreases, until categorization differences across successive trials are no longer observable (*Figure 5A*).

The 'delta rule' and the 'hierarchical update with decay' both involve a constant learning rate implying that the parameter changes following each perceptual decision do not decrease as the experiment progresses (*Figure 5B–C*). Although both models were able to qualitatively reproduce the main result, namely that the rate of acoustic /aba/ categorized as 'ada' was higher immediately after a fused McGurk trial, the delta rule without decay did not provide a good fit. The 'hierarchical update with decay' provided the best explanation for the experimental results. Specifically, its advantage over the 'delta rule' is that the update decays across trials after a perceptual decision towards a less volatile representation of the category, providing an effective empirical prior (*Figure 5C*).

## Discussion

Speech sound categories are constantly revised as a function of the most recently presented stimuli (*Samuel and Kraljic, 2009*; *Kleinschmidt and Jaeger, 2015*; *Heald et al., 2017*). The proposed model provides a possible account for fast and transient changes in speech sound categories when

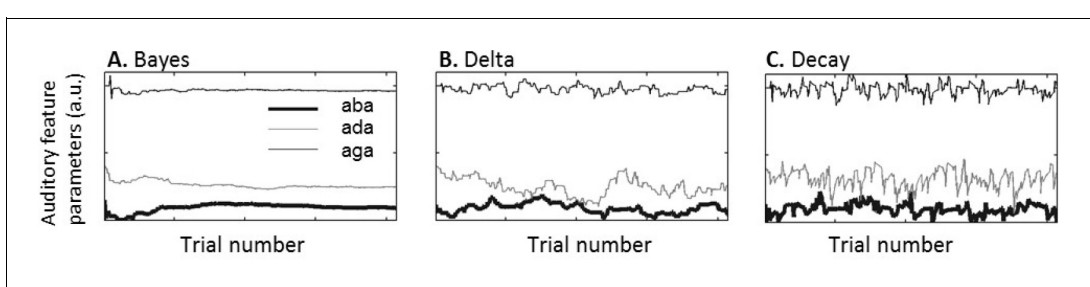

**Figure 5.** Category parameters across an experiment. $\theta_{/aba/,A}$, $\theta_{/ada/,A}$ and $\theta_{/aga/,A}$ after each of the 414 trials (69 repetitions of 6 different stimuli) in a sample simulated experimental run. (A) For the standard Bayesian model, category parameter updates become smaller as the experiment progresses. (B) For the constant delta rule updates of similar size occur throughout the experiment but are constant across trials. (C) Updates for the hierarchical delta rule with decay don't drift but decay to a long-term less volatile component.

confronted to a volatile sensory environment, involving the constant recalibration of internal models of speech.

The model was motivated by experimental results showing that /aba/ sounds were more frequently mis-categorized as 'ada' when they were preceded by a fused McGurk (*Lüttke et al., 2016a*). Since the reported effect was distinct from other well-documented across-trial dependency effects, such as perceptual priming or selective adaptation (*Heald et al., 2017*; *Gabay and Holt, 2018*), we sought to model it considering only changes in perceived speech categories without external feedback. Like previous approaches, ours builds on the idea that the brain achieves perception by inverting a generative model (e.g., *Rao and Ballard, 1999*; *Knill and Pouget, 2004*; *Friston, 2005*) and by continuously monitoring its performance to adapt it to changing stimulus landscapes. One way the brain can alter its internal models without external feedback is by using the perceptual outcome as a teaching signal or *ground truth* (*Luu and Stocker, 2018*) to induce model recalibration, such that the outcome better explains the sensory features that produced it.

Such auto-recalibration of speech sound categories has been described within an 'ideal adapter' framework (*Kleinschmidt and Jaeger, 2015*). In this framework, perceptual categories are subject to trial-by-trial changes well described by a Bayesian approach that implicitly assumes sound categories to be stable within an experimental session, and hence uses update/learning rules giving equal weight to recent and past evidence. While this makes sense in a stable environment where information remains equally relevant independent of its recency and the adaptation rule can describe the adaptation dynamics in sublexical speech categories in experiments with blocks of repeated stimuli (*Bertelson et al., 2003*; *Vroomen et al., 2007*), it failed here to account for the specific transient effect in *Lüttke et al. (2016b)*.

By comparing neurocomputational models with and without decay in their update rules, we show that recalibration occurs at least at two levels, characterized by two different decaying timescales: a fast decaying/short-term process (estimated to operate on the order of tens of seconds) that weighs recent versus immediate past sensory evidence, and a slow decaying/long-term process that keeps a longer trace of past sensory evidence to stabilize invariant representations. Such a dual process is critical because speech sounds have both variable (e.g. speakers) and stable components (e.g. phonemic categories).

The model was able to successfully reproduce transient speech category recalibration using empirically motivated update equations that include two hierarchically related parameters with different learning and decaying timescales. The fast changing variable is driven by current sensory prediction errors, and the more slowly changing one also driven by prediction errors but with a slower learning rate and hence acting as a longer-span buffer (*Figure 5*; *Equation (8)*). As a result, the parameter controlling the model expectations decays toward the longer-term parameter after a transient change driven by the current sensory prediction error. After a fused McGurk trial, the acoustic representation of /ada/ is shifted towards that of acoustic/aba/(*Figure 3A*), which increases the chances of an acoustic /aba/ to be mis-categorized as 'ada' (*Figure 3B*). Since the recalibration decays over time, the mis-categorization is most prominent for the acoustic /aba/ immediately following the McGurk fusion. On the other hand, because the decay drifts towards a more slowly changing 'version' of the /ada/ category, the model can also accommodate a more persistent accumulated adaptation as experimentally observed (*Bertelson et al., 2003*; *Vroomen et al., 2007*).

The model, which encompasses both the perceptual decision and the recalibration processes, is also consistent with activity observed in auditory cortex (*Lüttke et al., 2016a*). When the acoustic-only /aba/ was presented after a fused McGurk, fMRI activity in auditory cortex is more frequently classified as 'ada' by a learning algorithm trained to distinguish correctly identified acoustic /aba/, /ada/ and /aga/. The intermediate level in our perceptual model corresponds to the representation of acoustic features in auditory cortex and visual features in extrastriate visual cortex. In our model the amplitude of the 2nd formant transition (the acoustic feature that is most important for the aba/ada contrast) results from a combination of bottom-up sensory information and top-down predictions (*Equations 1, 2*). Since there are more /aba/ sounds perceived as 'ada' after a fused McGurk, the top-down predictions during the perceptual process are dominated by the /ada/ category, that is, our generative model predicts that activity in auditory cortex should be closer to /ada/ for acoustic /aba/ following fused McGurk trials, as shown by the fMRI data.

In summary, during audiovisual speech integration, and in McGurk stimuli in particular, the brain tends to find the most parsimonious account of the input, merging the acoustic and visual sensory

streams even at the expense of residual prediction errors at brain areas that encode unisensory stimulus features, represented in the model by the acoustic 2nd formant and lip amplitude modulation. Different participants may value differently this parsimony/accuracy trade-off. Those who consistently fuse the two streams presumably recalibrate their category representations (e.g. /ada/ after fused McGurk) thereby reducing residual prediction errors at the feature level. That is, we suggest that recalibration does not happen primarily at the areas encoding the stimulus features, but at higher order areas that encode sub-lexical speech categories, in a process that updates categories at least with two time scales.

## Non-Bayesian models of speech category learning and recalibration

As our interest lies at the computational level, we have not tested other, non-Bayesian models of speech category learning and recalibration (reviewed in *Heald et al., 2017*). Some existing models involve processes that are similar to ours; for example, non-Bayesian abstract models of new speech category learning use Gaussian distributions with parameters that are updated with delta rules similar to that of *Equation 7* (*Vallabha et al., 2007a*; *McMurray et al., 2009*). On the other hand, connectionist models of speech category learning posit a first layer of units with a topographic representation of the sound feature space and a second layer representing individual speech categories. Learning or recalibration is modelled by changing the connection weights between the two layers. In this way, *Mirman et al. (2006)*, modelled the recalibration of established prelexical categories that arises when an ambiguous sound is disambiguated by the lexical context as in the Ganong effect – a sound between /g/ and /k/ that tends to be classified as /g/ when preceding 'ift' or as /k/ when preceding 'iss' (*Ganong, 1980*). This effect shares some similarities with the McGurk effect, although the latter is stronger as it changes the perception of a non-ambiguous sound. The benefit of having two timescales is also illustrated by a connectionist model of the acquisition of non-native speech sound categories in the presence of well-established native ones (*Vallabha and McClelland, 2007b*). Interference between new and existing categories was avoided by positing a fast learning pathway applied to the novel categories, and a slower learning pathway to the native ones. Finally, connectionist models can also reproduce short-term effects such as perceptual bias and habituation (*Lancia and Winter, 2013*). These examples suggest that a connectionist model could provide a physiologically plausible instantiation of our abstract Bayesian model as long as one incorporates two pathways with two different timescales or a single pathway that uses metastable synapses (*Benna and Fusi, 2016*).

## Advantages of a dual time-scale representation

Parallel learning systems working at different temporal scales have previously been proposed in relation to speech; one able to produce fast mappings and heavily relying on working memory, while the other relies on procedural learning structures that eventually results in effortless, implicit, associations (*Myers and Mesite, 2014*; *Zeithamova et al., 2008*; *Maddox and Chandrasekaran, 2014*). Our proposal can theoretically be motivated on similar grounds. We argue that the brain implements at least two representations of natural categories; one more flexible than the other. The more flexible one might be used to achieve the agent's current goal, while the more stable and less precise representation keeps general knowledge about sound categories. We propose the term 'working' representation for the more flexible sound category representation, to distinguish it from its more stable 'episodic' form.

Behaviourally, this can be advantageous when specific instances of a unique category, for example from a single speaker, have less associated uncertainty than the overall prior distribution across all possible instances across speakers. The 'working representation' corresponds to an 'intermediate' representation that has lower uncertainty and therefore makes the sensory integration process more precise, leading to more confident perceptual decisions at the single trial level. This strategy allows the agent to use a precise 'working' category that can quickly change from trial to trial.

In this view, the agent needs to infer the distribution (mean and covariance) that defines the working representation, and combine sensory evidence with the prior, that is, with the corresponding long-term 'episodic' representation. Based on these Bayesian principles we wrote the hierarchical recalibration rule (*Equation 8*), which appears whenever there are three quantities informing the current estimate of a variable (under Gaussian conditions). In our model, the current expected value for

the working representation is informed by the value derived from the observation in the previous and current trials, and from the episodic representation as schematized in the right panel of *Figure 6*. Bayesian inference assuming known volatilities for the two levels in *Figure 6*, and under the mean-field approximation, can be calculated analytically resulting in update equations that take the form written at the bottom of the diagram. The coefficients, which determine what is referred to as 'learning rates' in the reinforcement learning literature, are functions of the parameters of the model related to the different sources of uncertainty: volatilities at both hierarchical levels, sensory noise and the width of the episodic representation. Our proposed update equation therefore assumes that agents have already estimated the volatilities at the two levels (as in *Behrens et al., 2007*; *Nassar et al., 2010*; *Mathys et al., 2014*).

Finally, from the optimal agent's perspective, the internal model used for a given trial is the predictive working representation built from 1) updated representations after the last observation and 2) their expected change in the intervening time. The latter component denotes the uncertainty associated with the representations, for example more volatile representations becoming more uncertain more quickly. This last point is important as it means that, across trials, increased uncertainty associated with the previous estimate of the working representation implies more reliance on the long-term representation. In other words, across trials, the expected sound feature modulation encoded by the working representation 'decays' back towards that of the long-term representation. This reflects a form of 'optimal forgetting', that is, the expected loss of relevance of a past observation for the current trial.

## Functional neuroanatomy of transient category shifts

Whether the two time scales that were needed to explain simultaneous tracking of long and short-term category representations are hierarchically organized or implemented in parallel, and what brain regions or mechanisms might be implicated is an open question.

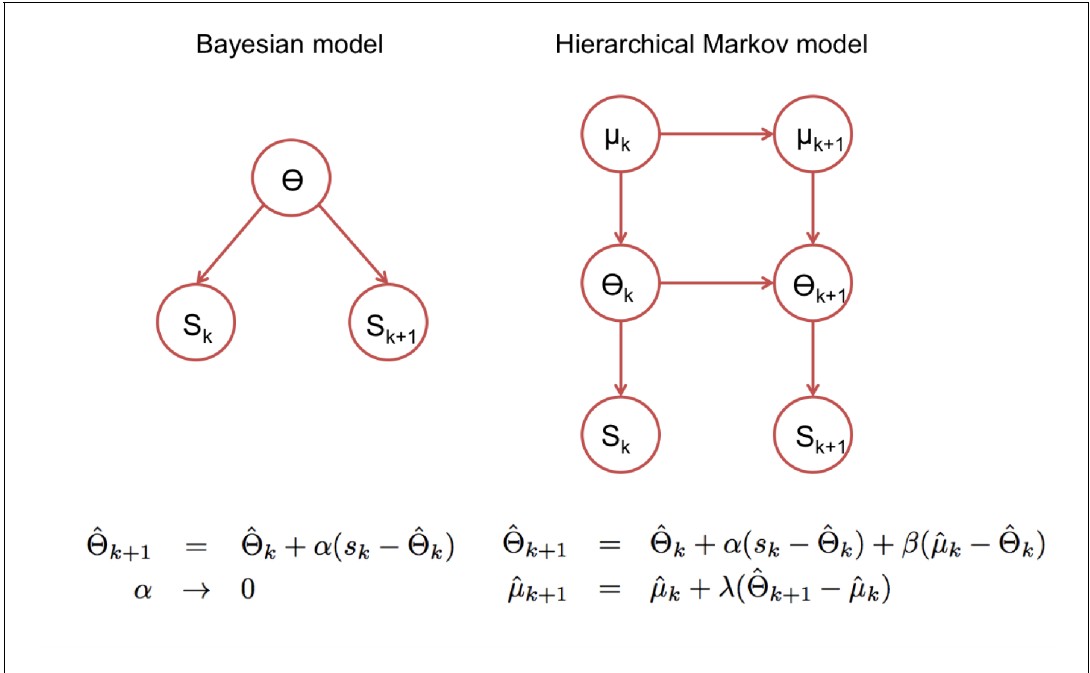

**Figure 6.** Statistical models underlying the two classes of update rules used in the paper. Here $\Theta$ stands for the model parameters that determine the speech categories used by the perceptual model (See *Figure 3*) and 'k' for the trial index. On the standard Bayesian approach (left), model parameters are considered constant in time leading to update rules that give the same weight to all prediction errors, which in turn leads to a 'learning rate' $\alpha_k$ that becomes smaller with the number of trials. On the right, we show a hierarchical Markov model implementation that would lead to the kind of update rules that we introduced empirically to accommodate the rapid recalibration effect. This alternative view implicitly assumes that model parameters can change in time and therefore lead to update rules with learning rates ($\alpha$, $\beta$ and $\lambda$) that under certain assumptions, settle to non-zero constant values.

There are at least three scenarios that could support recalibration at different time scales. First, a hierarchy of time scales might exist at the single synapse level. Hierarchically related variables with increasing time scales (dynamics described by equations as in *Figure 6B*) have been used in a modelling study to increase the capacity of memory systems and improve the stability of synaptic modifications (*Benna and Fusi, 2016*), and a model of synapses with a cascade of metastable states with increasing stability was able to learn more flexibly under uncertainty (*Iigaya, 2016*). In the latter model, the intrinsic decay of synaptic modifications was faster for the more labile memory states, i.e. those that are more sensitive to new evidence. In contrast, the deeper, more stable memory states, showed slower decay, which overall nicely concurs with our proposal. It is thereforepossible that the representations are encoded at synapses with the transient recalibration corresponding to synaptic modification at the more labile states and the long-term component residing in the less labile states.

A second option involves prefrontal cortex working in tandem with other brain regions. In perceptual classification tasks under volatile conditions, prefrontal cortex can flexibly combine alternative strategies, such as optimal Bayesian-like learning in stable environments and a working memory model in volatile environments (*Summerfield et al., 2011*). In our setting, to guide the speech classification process, it could conceivably combine a 'working' short time scale representation of the speech categories with a long-term 'episodic' representation, which might reside in different brain networks. Several fronto-parietal regions have indeed been implicated in controlling the effect of sensory and choice history on perceptual decisions: 'Sensory evidence, choice and outcome' could be decoded from ventrolateral prefrontal cortex and predicted choice biases (*Tsunada et al., 2019*). Neuronal responses in fronto-parietal circuits could provide a basis for flexible timescales (*Scott et al., 2017*), as dissociated effects of working memory and past sensory history have been found to involve the prefrontal cortex and posterior parietal cortices respectively (*Akrami et al., 2018*). The observed sensitivity to sensory choice history and sensory evidence is consistent with our model, which uses internal category representations to interpret sensory evidence, with category representations being recalibrated based on choice (i.e., the perceived category).

Finally, the hierarchical nature of perception and action (*Kiebel et al., 2008*; *Friston, 2008*) might be paralleled in the brain, by hierarchical processing in prefrontal cortex (*Badre, 2008*; *Koechlin and Jubault, 2006*; *Summerfield et al., 2006*) and sensory areas (*Felleman and Van Essen, 1991*; *Chevillet et al., 2011*). It is hence conceivable that the relation between the two timescales is hierarchical with higher-level representations becoming increasingly abstract and time insensitive. This could happen, for example, if the brain used representations at a speaker level that are drawn from more general representations of speech categories at the population level. Recalibration might work at every level of the temporal hierarchy, with higher levels integrating update information within increasingly longer time windows (longer timescales), making them less and less sensitive to new observations.

## Revised 'ideal adapter'

While the 'ideal adapter' account focused on cumulative recalibration (*Kleinschmidt and Jaeger, 2015*), our results suggest that shorter-lived effects are also behaviorally relevant. The ideal adapter could be formalized as a simple incremental optimal Bayesian inference in a non-volatile environment (*Figure 6*, left panel), whereas our update rule could be cast in a normative framework that explicitly accounts for environmental volatility. A hierarchical model with constant volatility at two levels (*Figure 6*, right panel) could lead to hierarchical update equations (*Wilson et al., 2013*) that can be approximated by constant learning rates (with higher learning rates - faster 'forgetting'- being related to stronger volatility). The right panel of *Figure 6* assumes that the higher level ($\mu$) has lower volatility than the intermediate level ($\theta$), hence combining volatility with hierarchy. This combination departs from models used to explain decision making in changing environments (*Behrens et al., 2007*; *Summerfield et al., 2011*; *Mathys et al., 2014*), which are not hierarchical, and focus on the nontrivial task of inferring the environment volatility. These studies show that human participants adapt their learning rates to the changing volatility, which could be modelled without keeping representations across several time scales. In these tasks, participants need to keep track of short-lived changes in arbitrary cue-reward associations or in arbitrarily defined sensory categories (*Summerfield et al., 2011*), whereas we model overlearned and behaviourally relevant categories, which also requires to maintain long-term estimates as empirical priors.

### Relevance to speech and language pathologies

Our modelling results are relevant to continuous speech processing, in particular to account for auditory processing anomalies in dyslexia. Evidence from a two-tone frequency discrimination task suggests that participants' choices are driven not only by the tones presented at a given trial, but also by the recent history of tone frequencies in the experiment, with recent tones having more weight than earlier ones (*Jaffe-Dax et al., 2017*). It turns out that, when compared with controls, subjects with dyslexia show a decreased reliance on temporally distant tones, suggesting a shorter time constant (*Jaffe-Dax et al., 2017*). Translating this result to the current model, we could hypothesize that in dyslexia the long-term component (μ in *Figure 6B*) has either a shorter time span, or is coupled to the lower representation with a lower weight. In both cases, we would expect a deficit in building long-term stable speech category representations since they would be overly driven by the current context. ASD individuals on the other hand, are optimally biased by long-term tones, but do not show the bias by short-term tones of neurotypical participants (*Lieder et al., 2019*), which suggests a faster decay or an absence of the short-term component in *Figure 6B*. This would predict a failure of ASD individuals to show the specific effect after McGurk trials in the experiment simulated here.

### Conclusion

We present a revised 'ideal adapter' model for speech sound recalibration that has both transient and cumulative components organized hierarchically. This new model provides evidence for a hierarchy of processes in the recalibration of speech categories, and highlights that after experiencing the McGurk effect, it is not the acoustic features related to the sensory input that are modified, but higher-level syllabic representations. The model implies that the activity changes in sensory cortices are not locally generated but reflect the interaction of bottom-up peripheral sensory inputs and top-down expectations from regions where categorical perception takes place. Considering natural speech processing as the inversion of a continuously monitored and recalibrated internal model can unveil the potential operations and strategies that listeners use when they are confronted with the acoustic volatility associated with speech categories, which by their nature have both rapidly changing (e.g. speaker specific) and slowly changing (e.g. speaker general) components. Such a model can be implemented by a hierarchy of empirical priors that are subject to changes at different time scales. Although developed in the context of speech processing, our proposal may also apply to other cognitive domains requiring perhaps more nested timescales, such as action planning (*Badre, 2008*; *Koechlin and Summerfield, 2007*; *Koechlin et al., 2003*).

## Materials and methods

### Generative model

The goal of inference is to establish which is the speech token that gave rise to the incoming sensory input. We restrict ourselves to three possible tokens: /aba/, /ada/ and /aga/ (as in *Lüttke et al., 2016a*). Although several acoustic and visual features can distinguish between them, we choose to model the 2$^{nd}$ formant transition, which is minimal for /aba/ but increases for /ada/ and /aga/, and the degree of lip closure, which is maximal for /aba/ and less prominent for /ada/ and /aga/(lip closure /aba/>/ada/>/aga/). This choice is based on the fact that what distinguishes between the three speech sounds is the place of articulation. Acoustically the 2$^{nd}$ formant transition is an important cue for place of articulation, particularly within the 'a' vowel context (*Liberman et al., 1957*); visually it is the degree of lip aperture at the time of the vocal cavity occlusion depending on its location (complete lip closure for the bilabial (/aba/), and decreasing lip closure for the alveolar (/ada/) and velar (/aga/) (*Campbell, 2008*; *Varnet et al., 2013*).

The generative model has three levels; the higher level encodes the speech token, the speech token in turn determines the expected values for the audiovisual cues, as represented in *Figure 1A*. The model includes the three possible tokens, each determining the expected distribution of its associated audiovisual features. We also introduce sensory noise to account for sensory variability. The parameters, location and spread of features associated with each token, as well as the parameter associated with the level of sensory noise, define an individual listener's internal model. That is, the listener models both the variability due to different articulations of the same speech category as well as the variability due to noise in the sensory system.

We use 'k' as the speech token index k = {/aba/,/ada/,/aga/}, 'f' as feature index f = {V,A}, where 'V' stands for the visual feature (lip aperture) and 'A' for the acoustic feature (2nd formant transition). The idea is that the amplitudes of the lip aperture and 2nd formant modulations vary according to the identity of the speech token ('k'). '$C_V$' or '$C_A$' denote hidden states associated with these amplitudes. Finally, '$s_V$' or '$s_A$' stand for the actual features in the audiovisual sensory input. The internal generative model considers '$s_V$' and '$s_A$' the versions of '$C_V$' or '$C_A$' corrupted by sensory noise ($\sigma_V$, $\sigma_A$).

There are two sources of variability, one related to sensory noise ($\sigma_V$, $\sigma_A$) and the variability of modulation amplitudes across different articulations of the same speech token ($\sigma_{k,V}$, $\sigma_{k,A}$), k = {/aba/ ,/ada/,/aga/}.

The hierarchical generative model is defined by the following relations:

$$p(s_f|C_f) \propto \exp\left(-\frac{(s_f - C_f)^2}{2\sigma_f^2}\right), f = A, V \tag{3}$$

$$p(C_f|k) \propto \exp\left(-\frac{(C_f - \theta_{k,f})^2}{2\sigma_{k,f}^2}\right), f = A, V, \ k = \text{speech token} \tag{4}$$

While the above defines the generative (top-down model) p($s_f$|$C_f$,k), our interest lies in its inversion p(k|$s_A$,$s_V$), where $s_A$,$s_V$ represents the sensory input in a single trial.

From the inversion of the model defined by the relations above one obtains:

$$p(k|s_A, s_V) \propto \exp\left(-\frac{(s_V - \theta_{k,V})^2}{2(\sigma_V^2 + \sigma_{k,V}^2)} - \frac{(s_A - \theta_{k,A})^2}{2(\sigma_A^2 + \sigma_{k,A}^2)}\right) p(k) \tag{5}$$

which results from marginalizing over '$C_V$' and '$C_A$'-the intermediate stages that encode the visual and acoustic features, explicitly:

$$p(k|s_A, s_V) = \int dC_V dC_A \, p(k, C_V, C_A|s_A, s_V) = \int dC_V dC_A \, p(s_V|C_V)p(C_V|k)p(s_A|C_A)p(C_A|k)p(k)$$
$$\propto \exp\left(-\frac{(s_V - \theta_{k,V})^2}{2(\sigma_V^2 + \sigma_{k,V}^2)} - \frac{(s_A - \theta_{k,A})^2}{2(\sigma_A^2 + \sigma_{k,A}^2)}\right)p(k)$$

$$\int dC_V \frac{1}{\sigma_{k,V}}\exp\left(-\frac{(C_V - C_{V,k})^2}{2\sigma_{V,k}^2}\right) \int dC_A \frac{1}{\sigma_{k,A}}\exp\left(-\frac{(C_A - C_{A,k})^2}{2\sigma_{A,k}^2}\right)$$
$$\frac{C_{f,k}}{\sigma_{f,k}^2} \equiv \frac{s_f}{\sigma_f^2} + \frac{\theta_{k,f}}{\sigma_{k,f}^2}, \frac{1}{\sigma_{f,k}^2} \equiv \frac{1}{\sigma_f^2} + \frac{1}{\sigma_{k,f}^2}, \ f = V, A$$

We assume that initial variance and prior probabilities are equal across categories (p(k)=1/3).

Alternatively, marginalization over 'k' gives the probabilities over the hidden variables '$C_V$' and '$C_A$', which we associate with encoding of stimulus features (lip aperture, 2nd formant transition) in visual and auditory cortex respectively.

$$p(C_V, C_A|s_V, s_A) = \sum_k p(k|s_V, s_A) \frac{1}{\sqrt{2\pi}\sigma_{V,k}}\exp\left(-\frac{(C_V - C_{V,k})^2}{2\sigma_{V,k}^2}\right) \frac{1}{\sqrt{2\pi}\sigma_{A,k}}\exp\left(-\frac{(C_A - C_{A,k})^2}{2\sigma_{A,k}^2}\right)$$
$$\frac{C_{f,k}}{\sigma_{f,k}^2} \equiv \frac{s_f}{\sigma_f^2} + \frac{\theta_{k,f}}{\sigma_{k,f}^2}, \frac{1}{\sigma_{f,k}^2} \equiv \frac{1}{\sigma_f^2} + \frac{1}{\sigma_{k,f}^2}, \ f = V, A$$

This shows explicitly how internal estimates of sensory features (lip aperture '$C_V$' and 2nd formant '$C_A$') are driven by bottom up sensory evidence ($s_V$, $s_A$) and top-down expectations related with each category 'k' contributing according to its internal expectations ($\theta_{k,V}$ $\theta_{k,A}$). When there is strong evidence for a given category 'k', the sum can be approximated by a single Gaussian centered at a compromise between $\theta_{k,V}$ and $s_V$ for the visual feature and between $\theta_{k,A}$ and $s_A$ for the acoustic feature.

In principle the model could also be made to perform causal inference (*Magnotti and Beauchamp, 2017*), that is, decide whether the two sensory streams belong to the same source and therefore should be integrated, or whether the two sensory streams do not belong to the same source, in which case the participant should ignore the visual stream. Since Lüttke et al. explicitly selected the participants that consistently reported /ada/ for the McGurk stimulus, these subjects

were fusing the two streams. We hence assume that integration is happening at every audio-visual trial.

The model's percept corresponds to the category that maximizes the posterior distribution: p(k| s$_A$, s$_V$).

## Recalibration model

The previous section presented how the model does inference in a single trial. We now turn to how the model updates the parameters that encode the internal representation of the three speech categories. This happens after every trial, thus simulating an internal model that continuously tries to minimize the difference between its predictions and the actual observations; we assume that in this process, in which the model tries to make itself more consistent with the input just received, it will only update the category corresponding to its choice. We will present three updating rules. The normative incremental Bayesian update model used by *Kleinschmidt and Jaeger (2015)*, and the empirically motivated constant delta rule and hierarchical delta rule with intrinsic decay.

## Bayesian updating

The internal representation of the speech categories is determined by six location parameters ($\theta_{k,V}$ $\theta_{k,A}$) and six width parameters ($\sigma_{k,V}$, $\sigma_{k,A}$). We follow *Gelman et al. (2003)* and define the following prior distributions for the internal model parameters ($\theta_{k,V}$, $\sigma_{k,V}$; $\theta_{k,A}$, $\sigma_{k,A}$). For each of the six ($\theta$, $\sigma$) pairs (2 sensory features × 3 categories) the prior is written as:

$$p(\theta_{k,f}, \sigma_{k,f}) = p(\theta_{k,f}|\sigma_{k,f})p(\sigma_{k,f})$$
$$\propto \sigma_{k,f}^2 \left(\sigma_{k,f}^2\right)^{-(v_{k,f,0}/2+1)} \exp\left(-\frac{[\kappa_{k,f,0}(\theta_{k,f}-\mu_{k,f,0})^2 + v_{k,f,0}\sigma_{k,f,0}^2]}{2\sigma_{k,f}^2}\right)$$

As above, f refers to the sensory feature, either V or A, and k to the speech category, either /aba/ , /ada/ or /aga/.

After a new trial with sensory input (s$_V$, s$_A$) the updated prior has the following parameters:

$$\kappa_{k,f,0} \leftarrow \kappa_{k,f,0} + 1$$

$$\mu_{k,f,0} \leftarrow \mu_{k,f,0} + \frac{1}{\kappa_{k,f,0}+1}(s_f - \mu_{k,f,0})$$

$$v_{k,f,0} \leftarrow v_{k,f,0} + 1$$

$$v_{k,f,0}\sigma_{k,f,0}^2 \leftarrow v_{k,f,0}\sigma_{k,f,0}^2 + \frac{\kappa_{k,f,0}}{\kappa_{k,f,0}+1}(s_f - \mu_{k,f,0})^2$$

After each trial the inference process described in the previous section determines the percept from the posterior probability p(k|s$_V$ s$_A$). Only the feature parameters of the representation corresponding to the percept are subsequently updated.

We use the values that maximize the posterior over the parameters given the input and the current estimated category 'k' to determine the point estimates that will define the updated model parameters for the next trial (*Equation 6*). The updates for the location and spread parameters then take the form:

$$\Delta\theta_{k,f} = \frac{1}{\kappa_{k,f,0}+n(k)}(s_f - \theta_{k,f})$$
$$\Delta\sigma_{k,f}^2 = \frac{1}{v_{k,f,0}+n(k)+1}\left[\frac{\kappa_{k,f,0}+n(k)-1}{\kappa_{k,f,0}+n(k)}(s_f - \theta_{k,f})^2 - \sigma_{k,f}^2\right]$$

(6)

Where 'k' is the perceived category, n(k) the number of times the category has been perceived, f = V,A designates the sensory feature and $v_{k,f,0}$ and $\kappa_{k,f,0}$ are parameters from the prior distribution. The larger $v_{k,f,0}$ and $\kappa_{k,f,0}$ are, the more k 'perceptions' it takes for the parameter values of category 'k' to plateau but also the smaller the updates after each trial.

## Constant delta rule

The above update equations implicitly assume that the environment is stable and therefore updated parameters keep information from all previous trials. This is the result of the generative model, which did not include a model for environmental parameter changes. Introducing expectations about environmental changes led us to consider rules with constant learning rates. We restrict ourselves here to updates for the six location parameters ($\theta_{k,V}$ $\theta_{k,A}$) of *Equation 6*.

We first considered a constant delta rule scaled by the evidence in favor of the selected category $p(k|s_V, s_A)$,

$$\Delta\theta_{kf} = A(s_f - \theta_{kf})p(k|s_V, s_A) \tag{7}$$

As in the Bayesian case, updates accumulate without decay between trials. The main difference is that $\theta_{k,f}$ is driven more strongly by recent evidence than by past evidence, implicitly acknowledging the presence of volatility.

## Hierarchical delta rule with decay

Finally we consider updates that decay with time. We reasoned that the decay should be towards parameter estimates that are more stable, which we denote by $\mu_{k,f}$. We propose a hierarchical relation, with updates in $\mu_{k,f}$ being driven by $\theta_{k,f}$, while updates in $\theta_{k,f}$ are driven by sensory evidence. At each trial all categories ($k'$) decay toward their long-term estimates and only the perceived category ($k$) updates both $\mu_{k,f}$ and $\theta_{k,f}$:

$$\begin{aligned} \Delta\theta_{k'f} &= D(\mu_{k'f} - \theta_{k'f}) \\ \Delta\theta_{kf} &= R_1(s_f - \theta_{kf})p(k|s_V, s_A) \\ \Delta\mu_{kf} &= R_2(\theta_{kf} - \mu_{kf})p(k|s_V, s_A) \end{aligned} \tag{8}$$

The first equation reflects the decay while the last two equations apply to the perceived category '$k$'. '$D$', '$R_1$' and '$R_2$' are constant parameters. $R_2$ was set to zero since we do not expect the long-term component to change significantly within the experimental session.

## Model simulations

We simulate the experimental paradigm in *Lüttke et al. (2016a)*, in which human participants were asked about what they heard when presented with auditory syllables or auditory syllables accompanied with a video of the corresponding speaker's lip movements. There were six stimulus types: three acoustic only stimuli: /aba/, /ada/ and /aga/ and three audiovisual stimuli, congruent /aba/, congruent /ada/ and McGurk stimuli, that is, acoustic /aba/ accompanied by the video of a speaker articulating /aga/. Each stimulus type was presented 69 times to each participant. In the original experiment three different realizations of each of the six types were used. In our simulations we use a single realization per stimulus that is corrupted by sensory noise.

As described above, our model proposes that syllables are encoded in terms of the expected amplitudes and variances of audiovisual features. The expected amplitudes were taken from the mean values across 10 productions from a single male speaker (*Olasagasti et al., 2015*), the amplitudes were then normalized by dividing by the highest value for each feature resulting in the values: ($\theta_{/aba/,A} = 0.1$, $\theta_{/ada/,A} = 0.4$, $\theta_{/aga/,A} = 1$), ($\theta_{/aba/,V} = 1$, $\theta_{/ada/,V} = 0.6$, $\theta_{/aga/,V} = 0.37$).

For the other parameters defining the perceptual model, variances and sensory noise levels, we explored 20 different combinations. Five possible values for the pair ($\sigma_V$ $\sigma_A$): (0.1, 0.1), (0.12, 0.12), (0.12, 0.15), (0.15, 0.12), (0.15, 0.15). For each, we used four possible ($\sigma_{k,A}$ $\sigma_{k,V}$) pairs: (0.1, 0.1), (0.1, 0.2), (0.2, 0.1) and (0.2, 0.2). Parameters outside this range typically led to categorization accuracy worse than that from the participants in *Lüttke et al. (2016a)*.

For each of the 20 parameter sets defining the perceptual model, we tested a set of values for the parameters that define each recalibration model. Standard Bayes has two free parameters per category and feature: $\kappa_{k,f,0}$ and $\nu_{k,f,0}$ (*Equation 6*). We tested the same values for all categories and features and therefore we drop the '$k$' (category) and '$f$' (feature) subscripts; $\kappa_{k,f,0} = \kappa_0$, $\nu_{k,f,0} = \nu_0$. ($\kappa_0$, $\nu_0$) = (1, 5, 10) $\otimes$ (1, 5, 10) (where $\otimes$ denotes the tensor product).

For the constant delta rule, there is a single parameter per category and feature; we use the same for all categories but tested different values for different features. The learning rates for the

visual feature ($R_V$) and acoustic feature $R_A$ tested were (*Equation 7*) ($R_V$, $R_A$) = {(0.05, 0.1) $\otimes$ (0.02, 0.04, 0.06, 0.08), (0.1, 0.2) $\otimes$ (0.1, 0.2, 0.3, 0.4, 0.5, 0.6), (0.4, 0.5, 0. 6) $\otimes$ (0.4, 0.5, 0. 6), (0.7, 0.8) $\otimes$ (0.7, 0.8)}. In this case, we tested different values for visual and acoustic to increase the chances of the delta rule to reproduce the experimental results.

The hierarchical delta rule with decay has three parameters per category and feature (*Equation 8*). For all the simulation we set $R_{2,V}$ and $R_{2,A}$ = 0. For the other two parameters D and $R_1$ (common across categories and features) we tested 35 pairs: ($R_1$, D) = (0.05, 0.1, 0.12, 0.14, 0.16, 0.18, 0.2) $\otimes$ (0.05, 0.1, 0.2, 0.3, 0.4).

Although we did not perform an exhaustive exploration of parameter space, the ranges tested were determined by the informal observation that parameters outside the ranges tested led to excessive recalibration—too many /aba/s categorized as 'ada' overall.

The stimuli presented to the modelled participant corresponded to the expected acoustic and visual features of their internal model. Thus if the internal model for /aba/ is centered at $\theta_{/aba/,A}$ for the acoustic feature and at $\theta_{/aba/,V}$ for the visual feature, those are the amplitudes chosen for the input stimuli. In other words, stimuli were tailored to the modelled participant. It is worth emphasizing that the modelled agent does not have a fused 'McGurk' category; their model only includes congruent expectations.

The six stimuli were defined by:

- Acoustic /aba/: ($C_V$ $C_A$) = ( $\emptyset$, $\theta_{/aba/,A}$)
- Acoustic /ada/: ($C_V$ $C_A$) = ( $\emptyset$, $\theta_{/ada/,A}$)
- Acoustic /aga/: ($C_V$ $C_A$) = ( $\emptyset$, $\theta_{/aga/,A}$)
- Congruent /aba/: ($C_V$ $C_A$) = ($\theta_{/aba/,V}$ $\theta_{/aba/,A}$)
- McGurk: ($C_V$ $C_A$) = ($\theta_{/aga/,V}$ $\theta_{/aba/,A}$)
- Congruent /aga/: ($C_V$ $C_A$) = ($\theta_{/aga/,V}$ $\theta_{/aga/,A}$)

Even if the underlying parameters for a given stimulus type were the same for every trial, sensory noise created variability. The input to the model was the pair $s_V$, $s_A$ defined by:

$$s_V = C_V + \sigma_V \eta_V$$
$$s_A = C_A + \sigma_A \eta_A$$

where $\eta_V$ and $\eta_A$ are sampled from independent Gaussian distributions with zero mean and unit variance. That is, $s_V$ $s_A$ are noisy versions of the true amplitude modulations in the visual and auditory modality.

The six stimulus types were presented to the model in random order with 69 repetitions for each. At the end of the presentation, the model chose a percept based on the posterior distribution over syllable identity 'k' given the stimulus. The perceived syllable was then recalibrated by updating its defining parameters (either both mean and variance or mean alone depending on the specific update rule).

In the model recalibration step, sometimes $\theta_{/ada/,A}$ became smaller than $\theta_{/aba/,A}$. This happened mostly for the constant delta rule as we increased the learning rate parameter, which also lead to increases in the McGurk contrast. Despite this modification, the observed McGurk contrast for the constant delta rule was not statistically significant. $\theta_{/ada/,A}$ becoming smaller than $\theta_{/aba/,A}$ constitutes a reversal of the initial relation between these parameters; empirically one finds that 2nd formant modulation is larger for /ada/ than for /aba/ ($\theta_{/ada/,A} > \theta_{/aba/,A}$). We included a line in our code that made sure that this did not occur. If after recalibration $\theta_{/ada/,A}$ was smaller than $\theta_{/aba/,A}$, the two were interchanged. This can be interpreted as a prior that incorporates information about the relations between categories. If reversals were accepted, subsequent acoustic /aba/ would be systematically classified as /ada/.

## Update rule evaluation

To evaluate the performance of the models, we used the following data from the original experiment. 1) The McGurk contrast, defined by two values: $p_{ada,Mc}$ the proportion of acoustic only /aba/ categorized as 'ada' when preceded by a fused McGurk; (29%) or by other stimuli $p_{ada,oth}$, acoustic /aba/ and /aga/ and congruent /aba/ and /aga/, (16%). 2) Overall performance: the proportion of the most frequent category for each of the six stimulus types: 80% of 'aba' percepts for acoustic only /aba/, 83% of 'ada' percepts for acoustic only /ada/, 98% of 'aga' percepts for acoustic only /aga/;

98% of 'aba' percepts for congruent audiovisual /aba/, 87% of 'ada' percepts for incongruent McGurk (acoustic /aba/ and visual /aga/), and 98% of 'aga' percepts for congruent /aga/. We will represent these values as the six entries of the vector $c_{stim}$.

While the original experiment had 27 participants, we run the experiment 100 times. By drawing 6000 random samples of 27 from the 100 runs, we estimated appropriate sampling distributions. For the quantities of interest listed in the previous paragraph we calculated the medians over the 6000 samples.

In a first step, for each update rule, we selected the model with the parameters that lead to the minimum mean squared error for the McGurk contrast $2\Delta^2_{Mc} = (p_{ada,Mc} - 29)^2 + (p_{ada,oth} - 16)^2$. The coefficients in the update rules appearing in the Results section correspond to those that lead to the minimum $\Delta_{Mc}$ for each update rule.

In a second step, we concentrated on the model with the best parameters for each update rule. The 6000 random samples of size 27 were used to build a sampling distribution for $\Delta_{Mc}$ and a measure of overall performance also based on a mean squared error: $6\Delta^2_{overall} = (c_{Ab} - 80)^2 + (c_{Ad} - 83)^2 + (c_{Ag} - 98)^2 + (c_{VbAb} - 98)^2 + (c_{McGurk} - 87)^2 + (c_{VgAg} - 98)^2$. Additionaly we calculated the 95% confidence intervals for the size of the McGurk contrast (the difference $p_{ada,Mc} - p_{ada,oth}$). We choose as representative for the 6000 samples, the sample with the median value for $6\Delta^2_{overall} + 2\Delta^2_{Mc}$.

For each of the 6000 random samples we tested whether the McGurk contrast paired values $p_{ada,Mc}$ and $p_{ada,oth}$ were significantly different (Wilcoxon signed rank test). In the results section we report the 95% confidence intervals for the difference $p_{ada,Mc} - p_{ada,oth}$, as well as the Wilcoxon signed rank test for the sample with the median value for $6\Delta^2_{overall} + 2\Delta^2_{Mc}$, (the representative sample from the 6000).

*Figure 4*,includes the control or '/ada/ contrast' for the representative sample. This contrast is defined by the proportion of acoustic only /aba/ sounds categorized as 'ada' when preceded by an acoustic only /ada/ correctly categorized as 'ada' or by other stimuli (acoustic only /aba/ or /aga/).

All simulations and statistical tests were performed using custom scripts written in MATLAB (Release R2014b, The MathWorks, Inc, Natick, Massachusetts, United States). The original MATLAB scripts used to run the simulations are available online (https://gitlab.unige.ch/Miren.Olasagasti/recalibration-of-speech-categories; copy archived at https://github.com/elifesciences-publications/recalibration-of-speech-categories; *Olasagasti, 2020*).

## Additional information

### Funding

| Funder | Grant reference number | Author |
| --- | --- | --- |
| Swiss National Science Foundation | 320030B_182855 | Anne-Lise Giraud |

The funders had no role in study design, data collection and interpretation, or the decision to submit the work for publication.

### Author contributions

Itsaso Olasagasti, Conceptualization, Formal analysis, Investigation; Anne-Lise Giraud, Conceptualization

### Author ORCIDs

Itsaso Olasagasti (iD) https://orcid.org/0000-0002-5172-5373

### Decision letter and Author response

Decision letter https://doi.org/10.7554/eLife.44516.sa1
Author response https://doi.org/10.7554/eLife.44516.sa2

## Additional files

### Supplementary files
- Transparent reporting form

### Data availability

The original MATLAB scripts used to run the simulations are available online (https://gitlab.unige.ch/Miren.Olasagasti/recalibration-of-speech-categories; copy archived at https://github.com/elifesciences-publications/recalibration-of-speech-categories).

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
