## [Decision Letter]

**Acceptance summary:**

In the real world, the acoustic environment changes continuously -- background noise can be loud or quiet, and even at a particular volume the spectrum can change dramatically (both on relatively short timescales). This work addresses the question of how speech perception adapts to such changes. The authors propose a model that is successful in explaining empirical data, including very recent data describing perceptual adaptation at multiple time scales. This should make an important contribution to our understanding of multi-scale speech processing.

**Decision letter after peer review:**

Thank you for submitting your article "Integrating prediction errors at two time scales permits rapid recalibration of speech sound categories" for consideration by *eLife*. Your article has been reviewed by two peer reviewers, and the evaluation has been overseen by a Reviewing Editor and Timothy Behrens as the Senior Editor. The reviewers have opted to remain anonymous.

The reviewers have discussed the reviews with one another and the Reviewing Editor has drafted this decision to help you prepare a revised submission.

Summary:

Olasagasti and Giraud constructed probabilistic models to study speech recalibration -- how speech perception adapts to changes in the acoustic environment of the perceiver. Unlike previous work, the current work addresses a volatile environment. The resulting model is successful in explaining empirical data, including very recent data describing perceptual adaptation at multiple time scales; in particular, an updating rule at two timescales outperformed models with an updating rule at a single scale. The work is important, the paper is well-written and the results will be of great value to the field of speech perception, and likely also to the field of perception in general.

Essential revisions:

1) The authors highlighted that updating rules at multiple timescales are beneficial, but it is probably the process implemented at each timescale that matters. I would like to ask the authors to further illustrate what is the nature of cognitive processes at each timescale, besides highlighting “two timescales”.

2) The sentence in the Abstract –“sound categories are represented at different time scales” – is not clear. Is it the information about sound category represented at different timescales? I would like the authors to clarify. The probabilistic models here represent a decision/inference process in my opinion, which is inconsistent with this claim. The frame-by-frame procedure of combining the visual/audio cues in the models are unrelated to the experimental evidence that humans can tolerate a large temporal lag between audio and visual cues of speech. I would like the authors to discuss the difference between the experimental evidence and their model procedures. If possible, could the authors jitter the lag between audio and visual cues to check the model performance. Could it be possible that the large timescale biased the model estimate in the beginning of each trial, even before perceptual information comes in. If possible, could the authors illustrate dynamics of the two- timescale model estimates as in Figure 2A.

3) The way in which the model is evaluated is not clear. Please be more specific about it – describe the relevant dataset, the evaluation of the model against the dataset and its comparison with other models in this respect.

4) Please elaborate on the principle of assigning different time scales to different levels in a hierarchical Bayesian inference framework; specifically, please describe how were specific time scales selected for specific Bayesian levels.

5) The advantages to speech perception are evident from this work, but the potential theoretical and computational consequences (difficulties, limitations and advantages) are not clear enough. Please devote a Discussion section to the broader aspect of evaluating time scales of adaptation.

---

## [Author Response]

Essential revisions:1) The authors highlighted that updating rules at multiple timescales are beneficial, but it is probably the process implemented at each timescale that matters. I would like to ask the authors to further illustrate what is the nature of cognitive processes at each timescale, besides highlighting “two timescales”.

In our approach, we have framed the recalibration process within the predictive/Bayesian brain framework that uses internal models to guide perception, in which the latter are continuously monitored and recalibrated such that they best explain sensory experience.

We use the term recalibration for the process by which the agent fine-tunes its model of the world, here representations of speech categories. This process should be considered as an advanced stage of learning that takes place after a category has been learned. It is known that, even at this point, categories are still relatively flexible.

It is important to note that learning depends on at least two systems. Fast changes are believed to rely on dopamine-dependent subcortical pathways that are dominant in the initial stages of learning a new category. As a category gets consolidated it is increasingly expressed through a cortical pathway that changes more slowly but also encodes information over long time spans.

Our shorter time scale could be related to the fast changing subcortical pathway and the longer time scale representation to that implemented through the consolidated cortical pathway.

When encountering a new speaker, the listener might use these two parallel processes with the less flexible cortical system encoding knowledge and expectations based on the entire past experience of the listener, and the more flexible system encoding changes over a short time scale.

The extent to which these two pathways are involved might be partly under cognitive control. The performance monitoring system might estimate volatility and flexibly determine how to weigh recent vs past experience.

As we elaborate in response to point 5 below, we propose that the “working” category representation meeting current goals is continuously being updated based on observations. From this perspective, the short time constant could be related to change detection and the long time constant to long-term consolidation of categories.

2) The sentence in the Abstract –“sound categories are represented at different time scales” – is not clear. Is it the information about sound category represented at different timescales? I would like the authors to clarify. The probabilistic models here represent a decision/inference process in my opinion, which is inconsistent with this claim. The frame-by-frame procedure of combining the visual/audio cues in the models are unrelated to the experimental evidence that humans can tolerate a large temporal lag between audio and visual cues of speech. I would like the authors to discuss the difference between the experimental evidence and their model procedures. If possible, could the authors jitter the lag between audio and visual cues to check the model performance. Could it be possible that the large timescale biased the model estimate in the beginning of each trial, even before perceptual information comes in. If possible, could the authors illustrate dynamics of the two- timescale model estimates as in Figure 2A.

We have modified the Abstract to clarify several points, including those raised here.

Our model consists of two steps, an inference/ perceptual decision step characterized by the frame-by-frame procedure illustrated in Figure 2A of the original submission, and a model recalibration step, taking place after the perceptual inference and decision step, in which the model is being updated. The internal model used in the decision/inference step is characterized by what we refer to as internal representations of the speech sound categories. Each category “representation” corresponds to a mapping between speech sound category and the expected stimulus features associated with that category; in this sense the categories “predict” sensory features. Our claim is that there are at least two such mappings with different timescales. They both determine the current belief of the agent. The mapping with the shorter timescale is the one used to derive the sensory feature estimates and the classification of the stimulus as “aba”, “ada” or “aga” at every trial (denoted with θ_k,f_ k: category, f: sensory feature). Once the stimulus has been categorized, both representations/mappings (θ_k,f_ and μ_k,f_) (information about sound category) are in principle updated. The difference is that the short time scale one (θ_k,f_) is driven more strongly by the estimated sensory features and also forgets them faster. The higher level representation with a longer timescale (μ_k,f_) does not change as much and also forgets less quickly and can therefore provide an empirical prior that works as an anchor for the representation at the lower-level with the shorter timescale. In our simulations we made the simplification that the recalibration of (μ_k,f_) was not observable in the course of the experiment.

Therefore, the two timescales do not refer to time constants that influence the inference process itself, and therefore do not relate to the issue of the temporal lag that humans can tolerate. The frame-by-frame inference process could work with any lag provided we explicitly include a lag variable and a prior on lag durations as in Magnotti et al., 2013 (Magnotti JF, Ma WJ, Beauchamp MS. Causal inference of asynchronous audiovisual speech. Front Psychol [Internet]. 2013 Jan [cited 2013 Dec 9];4:798). Moreover, lags could not be a factor in the current work because the experiment that we modelled did not vary lags.

Regarding the issue of whether the longer timescale biases the estimate at the beginning of the trial. As stressed in the previous paragraph, the two time-scales refer to how the parameters of the model are updated after the stimulus presentation, and keep information about past observations and categorizations. During the frame-by-frame process corresponding to the perceptual decision stage, the parameters are kept constant. However, they do “bias” the sensory feature estimates throughout the trial, in the sense that the estimates are linear combinations of the actual sensory features and the expected features from the category representations.

3) The way in which the model is evaluated is not clear. Please be more specific about it – describe the relevant dataset, the evaluation of the model against the dataset and its comparison with other models in this respect.

For this revision we have run more extensive simulations. In order to evaluate the models we used the average data and qualitative descriptions reported in (Lüttke 2015, 2016). This is now described in Materials and methods. Specifically, we used goodness of fit measures based on the McGurk contrast: proportion of acoustic aba stimuli miscategorised as “ada” when preceded by control stimuli (16%) or by fused McGurk stimuli (29%). We also used the reported overall performance of the participants: percentages of correctly identified acoustic “aba” (80%), correctly identified acoustic “ada” (83%), correctly identified acoustic “aga” (98%), correctly identified congruent audiovisual “aba” (98%), correctly identified congruent audiovisual “aga” (98%) and percentage of fused McGurk stimuli (87%).

In this round of simulations we varied parameters that define the perceptual model and parameters that defining the update rules. For each update method we chose the set of parameters that gave the best goodness of fit. We then compared the methods by comparing their 95% CI for the difference in the main McGurk contrast.

4) Please elaborate on the principle of assigning different time scales to different levels in a hierarchical Bayesian inference framework; specifically, please describe how were specific time scales selected for specific Bayesian levels.

In the setup represented in the right panel of Figure 6, time-scale is an emergent concept. As we elaborate in our answer to the next point, the learning rates and time scales are related to the coefficients in the update rules, which are in turn functions of the expected “volatility” at each level. The motivation to assign different expected volatilities lies on the fact that the causes for a change in how a speech category sounds in a given utterance by a given speaker can have different origins varying with time. For example, it could change because the speaker changes –a fast change- or because of a slower drift impeding on speech categories (e.g. deafness), which would not be measurable within an experimental session.

In our simulations we chose zero volatility – infinite time scale – for the longer-term representation. The timescale of the more volatile representation is a free parameter of the model.

In the present revision we have varied the assigned values indirectly by running simulations using different values for the coefficients of the update rule.

5) The advantages to speech perception are evident from this work, but the potential theoretical and computational consequences (difficulties, limitations and advantages) are not clear enough. Please devote a Discussion section to the broader aspect of evaluating time scales of adaptation.

Our model implies that the brain somehow incorporates at least two representations of natural categories, a flexible and volatile one, e.g. used for the agent’s current goals, and a more stable one that keeps general and less accurate knowledge about sound categories. We propose the term “working representation” for the more flexible sound category representation, to distinguish it from its more stable “episodic representation”. Eventually, the episodic representation itself could also have several subcomponents (not modelled here), which relates to the issue raised by the reviewers about the evaluation of the timescales of adaptation.

Behaviourally, this can be advantageous when specific instances of a unique category, for example from a single speaker, have less variance and therefore less associated uncertainty than the overall prior distribution over all possible instances. Having an “intermediate level” representation with lower uncertainty makes the sensory integration process more precise, leading to more confident perceptual decisions at the single trial level. In the case of speech, this strategy allows the agent to change its working category to a more precise one, while not forgetting the underlying distribution from which the current instance has been drawn.

If this is indeed what agents do, they need to infer the distribution (mean and variance) that defines the working representation, and according to Bayesian principles they should combine sensory evidence with the prior (in this case, the memory representation). Based on these Bayesian principles we wrote the update rule in Equation 8, that appears whenever there are three quantities informing the current estimate of a variable (under Gaussian conditions). In our model, the current expected value for the working representation is informed by the value derived from the observation in the previous and current trials, and from the episodic representation -- Figure 6, right diagram. Bayesian inference assuming known volatilities for the two levels in Figure 6, and under the mean-field approximation, can be calculated analytically resulting in update equations at the bottom. The coefficients, which determine what is referred to as “learning rates” in the reinforcement learning literature, are functions of the parameters of the model related to the different sources of uncertainty: the constant volatilities at both levels, sensory noise and the width of the memory representation.

Our proposed update equation therefore assumes that agents have estimated the volatilities. The trial-by-trial behavioural data of individual participants in Lüttke et al., 2016, if available, would help estimate the effective learning rates.

Finally, from the optimal agent’s perspective, the working representation used for a given trial is the predictive working representation built from 1) updated representations after the last observation and 2) their expected change in the intervening time. The latter component denotes the uncertainty associated with the representations, e.g. more volatile representations becoming more uncertain more quickly. This last point is important as it means that, across trials, increased uncertainty associated with the previous estimate of the working representation implies more reliance on the long-term representation. In other words, across trials, the working representation “decays” back towards the long-term representation. This reflects a form of “optimal forgetting” corresponding to the expected loss of relevance of a past observation to the current trial.

Many questions remain to be answered. Notably, is the fast time scale adapting to the current environment under “cognitive” control, or is it a default component in a system with a hierarchy of fixed timescales that can be flexibly combined?

Our work originated from our interest in speech perception, however, we think that it could apply to other categorization/abstraction processes that involve natural categories such as odors, colors or faces.

We have included some of these issues in the Discussion, particularly in the paragraph “Advantages of a dual time-scale representation.”.